# Quantitative Analysis of Core Lipid Production in *Methanothermobacter marburgensis* at Different Scales

**DOI:** 10.3390/bioengineering9040169

**Published:** 2022-04-10

**Authors:** Lydia M. F. Baumann, Ruth-Sophie Taubner, Kinga Oláh, Ann-Cathrin Rohrweber, Bernhard Schuster, Daniel Birgel, Simon K.-M. R. Rittmann

**Affiliations:** 1Institute for Geology, Center for Earth System Research and Sustainability, Universität Hamburg, Bundesstraße 55, 20146 Hamburg, Germany; lydia.feli.baumann@gmail.com (L.M.F.B.); geo@rohrweber.com (A.-C.R.); daniel.birgel@uni-hamburg.de (D.B.); 2Archaea Physiology & Biotechnology Group, Department of Functional and Evolutionary Ecology, Universität Wien, Djerassiplatz 1, 1030 Wien, Austria; ruth-sophie.taubner@univie.ac.at; 3Institute for Synthetic Bioarchitectures, Department of Nanobiotechnology, University of Natural Resources and Life Sciences, Muthgasse 11, 1190 Vienna, Austria; nagykinga012@gmail.com (K.O.); bernhard.schuster@boku.ac.at (B.S.); 4Institute for Chemical Technology of Organic Materials, Johannes Kepler Universität Linz, Altenbergerstraße 69, 4040 Linz, Austria; 5Arkeon GmbH, Technopark 1, 3430 Tulln an der Donau, Austria

**Keywords:** archaea biotechnology, microbial physiology, anaerobe, isoprenoids, ether lipids, methanogens

## Abstract

Archaeal lipids have a high biotechnological potential, caused by their high resistance to oxidative stress, extreme pH values and temperatures, as well as their ability to withstand phospholipases. Further, methanogens, a specific group of archaea, are already well-established in the field of biotechnology because of their ability to use carbon dioxide and molecular hydrogen or organic substrates. In this study, we show the potential of the model organism *Methanothermobacter marburgensis* to act both as a carbon dioxide based biological methane producer and as a potential supplier of archaeal lipids. Different cultivation settings were tested to gain an insight into the optimal conditions to produce specific core lipids. The study shows that up-scaling at a constant particle number (n/n = const.) seems to be a promising approach. Further optimizations regarding the length and number of the incubation periods and the ratio of the interaction area to the total liquid volume are necessary for scaling these settings for industrial purposes.

## 1. Introduction

The membrane lipids of archaea can be considered the most outstanding adaptations of life. Their unique lipid composition enabled archaea to conquer not only the mesophilic realm, but also the most extreme ecological niches on Earth, including those with high and low temperatures and pH values, high salinities, and anoxic environments (e.g., [1,2,3]). In contrast to the cell membranes of Bacteria and Eukarya, which are composed of ester-bound acyl chains at the *sn*-1 and *sn*-2 position, archaeal cell membranes are made of isoprenoid chains bound to glycerol by ether bonds at the *sn*-2 and *sn*-3 position (e.g., [4,5]). The generally high thermal and chemical stability of ether-bound, isoprenoidal archaeal membrane lipids, and the maintenance of their function at a wide range of chemical and physical conditions, make them a valuable study object and resource for biotechnology, biomedicine, and the pharmaceutical industry as, for example, liposomes or lipid films (e.g., [6,7]).

The membrane-spanning tetraether lipids, with their C_40_ alkyl chains connecting the two glycerol backbones, are of particular interest for the production of archaeosomes and lipid films (e.g., [6,7,8]). Among tetraether-lipid-producing archaea, methanogenic archaea, also referred to as methanogens (e.g., [9]), produce high amounts of these membrane lipids. Different methanogens produce various isoprenoidal hydrocarbons, di-, and tetraether lipids. Tetraether lipids are usually found as GDGTs (glycerol dialkyl glycerol tetraethers), GMGTs (glycerol monoalkyl glycerol tetraethers), and GTGTs (glycerol trialkyl glycerol tetraethers) (e.g., [4,9,10]), but are also known from non-methanogenic archaea (e.g., [11,12,13]). However, so far, only one methanogen has produced GDGTs and GMGTs with additional methyl groups [14], which makes them unique producers of these lipids.

Methanogens represent a group of archaea that require strict anaerobic conditions for optimal growth but are nevertheless widespread and occur, for example, in marine and lacustrine sediments; at hydrothermal vents, swamps, rice paddies, and soils; and in the gastrointestinal tracts of various animals, including humans (e.g., [15,16]). They are known to be able to metabolize a variety of gaseous substrates, e.g., carbon dioxide (CO_2_) and molecular hydrogen (H_2_) (e.g., [17]). Methanogens are widely used for anaerobic wastewater treatment and biogas production, whereas anaerobic digestors produce less sewage sludge—a costly by-product—than aerobic digestors (e.g., [18]). Besides the applications regarding their lipid inventory and wastewater treatment, currently, a considerable interest has emerged to employ methanogens for the conversion of CO_2_ to methane (CH_4_) using H_2_ (e.g., [19]). This would be a way to store carbon and sustainably produce and/or convert energy. One example concerns the CO_2_-based biological CH_4_ production (CO_2_-BMP) process, wherein autotrophic, hydrogenotrophic methanogens are utilized [19,20,21,22].

Specifically, one thermophilic methanogen, *Methanothermobacter marburgensis* [23,24,25], is currently the most promising model organism for CO_2_-BMP biotechnology. High rates of CO_2_ to CH_4_ conversion were obtained in a continuous culture with *M. marburgensis* [26]. During CO_2_-BMP, the conversion of CO_2_ and H_2_ to CH_4_ is performed according to the following stoichiometry:CO_2_ + 4 H_2_ → CH_4_ + 2 H_2_O   △G^′m^ = −126.6 ± 24.6 kJ mol^−1^(1)

Other characteristics of *M. marburgensis* make the strain even more intriguing for further physiological and biotechnological studies. The biochemistry of *M. marburgensis* is very well known [27]; it can be grown at very high specific growth rates up to 0.69 h^−1^ in minimal medium to high cell densities in bioreactors [21]; and the genome was sequenced [28]. Further, the physiological characteristics of the organism are well-known from closed batch [29], fed-batch [21], and continuous culture [22,23,26,30,31] experiments. *M. marburgensis* was chosen for this study, as it is one of the key organisms of Archaea Biotechnology [7,32].

The membrane core lipids of *M. marburgensis* have been described before [33] and comprise a range of isoprenoidal di- and tetraether lipids, partly with additional methylations in the alkyl chains (Figure 1). In contrast to intact polar lipids, the studied lipids do not contain the polar headgroups. None of the core lipids contain cyclopentane or cyclohexane moieties in the chains. This is indicated by a “0” after the name of the lipid. In this study, the nomenclature of Knappy *et al*. is used, which was originally introduced for a close relative of *M. marburgensis*, *Methanothermobacter thermautotrophicus* [14].

In general, archaeal lipids feature a higher resistance to oxidative stress and phospholipases, as well as a wide range of pH values and temperatures, compared to bacterial phospholipids [34,35]. Archaeal lipids can be used to manufacture artificial lipid films. Films made from archaeal lipids reveal low permeability, good insulating properties, and long-term stability [6,36]. Potential applications are in the fields of nanotechnology, biosensor design, and biomimetics [6]. Recently, catheter surfaces were coated with monolayers of tetraether lipids from the archaeon *Thermoplasma acidophilum* to avoid the adherence of pathogens [37]. Another application of archaeal lipids includes the production of liposomes. Liposomes are artificial lipid vesicles produced from phospholipids. They are tailored for their use in imaging diagnostics, as carriers of drugs, DNA, or peptides, and as adjuvants in vaccine therapy [8,38,39,40,41,42,43]. To date, liposomes were mostly made from ester phospholipids harvested from eukaryotes, such as egg phosphatidylcholine or hydrogenated soy phosphatidylcholine (e.g., [8,38,39,40,43]). Liposomes manufactured from archaeal lipids are referred to as archaeosomes (e.g., [8]). Compared to liposomes made from ester lipids, archaeosomes, especially tetraether-lipid-based archaeosomes, exhibit greater chemical and mechanical stability against very low and very high temperatures and pH, oxidative stress, lipases, bile salts, and serum media [6,8,38,39,40,44,45]. Archaeosomes were shown to have a higher stability in the gastrointestinal tract and to possess a longer shelf life (even in the presence of air or molecular oxygen); they can undergo heat sterilization and they showed no toxicity in mice [6,8,38,39,43]. Among other sources, archaeosomes can be produced using total polar lipid extractions from methanogens, e.g., from *Methanobrevibacter smithii* [7,46,47]. These archaeosomes showed an improved immune response in comparison to the response triggered by non-archaeal phospholipids. It could furthermore be shown that the long-lasting and robust immune response could be attributed to caldarchaeol, which acted as an adjuvant. The issue with these archaeosomes was that the batch-to-batch dependent composition of extracted *M. smithii* total polar lipids made it impossible to reproducibly generate archaeosomes with an identical lipid composition [7,48].

The aim of this study was to investigate how the growth conditions alter the quality and quantity of the membrane core lipids produced by *M. marburgensis*. The emphasis in the experiments was to investigate whether *M. marburgensis* varies the specific lipid production rate, product-to-product yield, and the quality of the core lipid composition under different environmental conditions. These conditions are potentially growth-limiting ratios of gas/liquid substrates (different volume ratios tested) and either a constant volume (V/V = const., at the starting point of the experiment) or a constant particle number (n/n = const., at the starting point of the experiment) of the gaseous substrate. Additionally, several experiments were conducted at a higher scale for comparison. The focus was to examine whether (a) the total amount of gaseous substrates or (b) the volume ratio of gas-to-liquid phase was the basis for the production of a particular type of core lipid. The experimental approach presented here allows insights into the physiological adaptability of the membrane lipids of *M. marburgensis* and the growth parameters crucial for their adaptations, and it provides a base for approaches to scale lipid production with this strain.

## 2. Materials and Methods

### 2.1. Archaeal Strain and Culture Set-Up

The thermophilic, hydrogenotrophic methanogen *Methanothermobacter marburgensis* DSM 2133^T^ was originally isolated from mesophilic sewage sludge [24]. The *M. marburgensis* culture used in this study was taken from our in-house methanogen strain collection (Archaea Physiology & Biotechnology Group, Department of Functional and Evolutionary Ecology, Universität Wien, Wien, Austria). *M. marburgensis* was originally obtained from the Deutsche Sammlung für Mikroorganismen und Zellkulturen GmbH (Braunschweig, Germany).

Cultures were grown either in 120 mL glass serum bottles (La-Pha-Pack, Langerwehe, Germany) or in 500 mL glass laboratory bottles (pressure plus+, narrow neck, with thread GL 45, DURAN^®^, DWK Life Sciences, Wertheim, Germany) with caps (screw cap with hole, PBT, red GL 45, Lactan, Graz, Austria). The 120 mL bottles have an empirically determined volumetric capacity of approximately 117 mL with inserted blue rubber stoppers (20 mm, butyl rubber, CLS-3409-14, Chemglass Life Sciences LCC, Vineland, NJ, USA). The larger glass bottles are marketed as 500 mL bottles. However, the actual volumetric capacity, which was determined empirically, and which was also used for calculations, is approximately 570 mL with inserted rubber stopper (black butyl rubber for GL45 bottles, Glasgerätebau Ochs, Bovenden/Lenglern, Germany).

The preparation, cultivation medium, inoculation, incubation, and harvesting described below were identical in the 117 mL and in the 570 mL bottles, except that the 570 mL bottles had shorter incubation intervals caused by the longer cooling down periods (see Section 2.4). For a better comparability of the experimental settings, the volumes were scaled up from the 117 mL experiments, with the final volume illustrated in Figure 2 (for the exact amount of media and inoculum, see Appendix A).

Quadruplicates of each volume and pressure variant were applied with an additional zero control to each experimental set for the 117 mL experiments. Due to logistic reasons, the experiments performed with the 570 mL flasks were performed in triplicates without zero controls. Inoculation was done in an anaerobic chamber (Coy Laboratory Products, Grass Lake, MI, USA) from a cultivated inoculum in defined media (Section 2.2). The residual CH_4_ was interchanged with a H_2_/CO_2_ (4:1 ratio, 99.995% purity (Air Liquide, Schwechat, Austria)) gas mixture twice per day with different pressure values adjusted to 1.1, 1.5, or 2 bar. The pressure is given as bar relative to atmospheric pressure throughout this study. The experimental conditions and settings are detailed in Figure 2 (and Appendix A), which shows an overview of the gas phase pressure values and the volumetric (V/V) and molar (n/n) constant alternatives. For the experiments termed as V/V = const, the initial liquid volume was constant with varying pressure (1.1, 1.5, 2.0 bar) within the 117 mL (25.3, 50.6, 75.9 mL) and 570 mL (123.3, 246.5, and 369.8 mL) settings, respectively. Note that the volumetric ratios between the liquid and gaseous phase in the 117 mL and 570 mL bottles are the same for the V/V = const. setting. In contrast, for the n/n = const. settings, the aim was to have the same initial number of moles of the gas phase in three different pressure settings (treated as ideal gas at room temperature, 22 °C). These three settings were defined by the number of moles at 1.5 bar in the 117 mL bottles, at 25.3, 50.6, and 75.9 mL initial volumes. The different starting volumes of the experiments performed at different pressure values were then calculated according to the ideal gas law. For better readability, we refer in further text to the small, medium, and large volumes in the 117 mL or 570 mL bottles at the conditions V/V = const. or n/n = const. The experiments in the 117 mL setting were conducted twice at two different incubation intervals and total times.

### 2.2. Cultivation Medium

The exact procedure for the medium preparation and the medium composition were as previously described [29]. The medium was aliquoted with regard to the proper volumes (Figure 2 and Appendix A) into 117 mL and 570 mL bottles and sealed with blue and black rubber stoppers, respectively, which were boiled ten times for 30 min in fresh ddH_2_O as a pretreatment. The 117 mL serum bottles were sealed with 20 mm aluminum crimp caps (Glasgerätebau Ochs, Bovenden/Lenglern, Germany). Anaerobization was ensured by gassing with a H_2_/CO_2_ (4:1 ratio) gas mixture (approximately 0.8 bar) five times and drawing vacuum four times. Afterwards, the bottles were autoclaved. In a final step, sterile 0.5 mol L^−1^ Na_2_S · 9H_2_O was added to the bottles in the anaerobic chamber (0.1 mL per 50 mL).

### 2.3. Incoulation

The inoculation was done using pre-cultures in an exponential growth phase. The exact ratios between medium and inoculum volume can be found in Appendix A, and the final total volume in Figure 2. The final steps of the preparation of the flasks were the inoculation performed in the anaerobic chamber and the final pressurization to approximately 1.1, 1.5, and 2.0 bar (±0.2 bar), respectively, with a H_2_/CO_2_ (4:1 ratio) gas mixture. The bottles were incubated in the dark in a shaking water bath at 65 ± 1 °C.

### 2.4. Pressure Measurement and Gassing

The 117 mL serum bottles were taken out and cooled down to room temperature before each pressure measurement (30 to 45 min). For the 570 mL bottles, the interval of the cooling-down stage increased due to the higher volume of the bottles, meaning on average 2.5 h; therefore, the incubation intervals resulted in a shorter period. After reaching room temperature, the bottle headspace pressure was measured using a digital manometer (LEO1-Ei, −1… 3 bar rel, Keller, Germany) with filters (sterile syringe filters, w/0.2c μm cellulose, 514-0061, VWR International, Wien, Austria), and cannulas (Gr 14, 0.60 × 30 mm, 23 G × 1 1/4”, RX129.1, Braun, Maria Enzersdorf, Austria). The gas phase of all bottles was exchanged with the H_2_/CO_2_ gas mixture detailed beforehand. The abovementioned routine took place twice per day (including the zero control bottles), and the bottles were incubated again in a water bath at 65 ± 1 °C (for details, see [29]). From the difference of the bottle headspace pressure before and after the incubation, the methane evolution rate (MER) was calculated (see Appendix A).

### 2.5. End Point OD Measurement and Harvesting

Subsequent to the last cooling-down stage and following the final pressure measurement with the digital manometer, a homogenous 0.7 mL sample was taken of each flask (117 and 570 mL) for end point optical density measurements (OD, λ = 578 nm, ddH_2_O serving as blank; used spectrophotometer: DU800, Beckman Coulter, Fullerton, CA, USA). Centrifugation was done in a pre-cooled centrifuge (4 °C, Heraeus Multifuge 4KR Centrifuge, Thermo Fisher Scientific, Osterode, Germany) at 4400 rpm for 20 min. The biomass (cell pellets) was transferred to 1.5 mL Eppendorf tubes. These Eppendorf tubes were further centrifuged at 4 °C for 15 min at 16,100 rpm (4 °C, Cooled Centrifuge 5424 R, Eppendorf AG, Hamburg, Germany), and the cell pellets were then stored at −20 °C until further analysis.

### 2.6. Lipid Extraction

The lyophilization was accomplished at −81 °C for 72 h (Alpha 2-4 LMC, Martin Christ Gefriertrocknungsanlagen GmbH, Osterode am Harz, Germany). The freeze-dried samples were aliquoted (1–20 mg) and homogenized with acetone-cleaned spatulas in glass centrifuge tubes (Präparatengläser Duran, 16 × 100 mm, PTFE-filled caps, Glasgerätebau Ochs, Bovenden/Lenglern, Germany). Then, 5 µg 5-α-cholestane (diluted from 10 mg mL^−1^ in chloroform, SUPELCO) and 5 µq DAGE C_18:18_ (dialkyl glycerol diether, 1,2-Di-O-octadecyl-rac-glycerol, Cayman Chemical, Biomol GmbH, Hamburg, Germany) were added as preparation standards. The samples subsequently underwent acid hydrolysis (2 mL of 10% (V/V) hydrochloric acid in methanol per sample) at 110 °C for 2 h. After that, core lipids were extracted four times with a mixture of n-hexane and dichloromethane (80:20) to obtain the total lipid extract (TLE). To an unfiltered, underivatized aliquot of each TLE, C_46_ GDGT [49] was added as an internal standard prior to injection into a Varian MS Workstation 6.91 High Performance Liquid Chromatography (HPLC) system coupled to a Varian 1200 L triple quadrupole mass spectrometer. The Atmospheric Pressure Chemical ionization (APCI) interface was operated in positive ion mode. Response factors varied and were carefully monitored (measurement of standard mixture after four sample measurements). Details about the measurements, temperature, and solvent program can be found elsewhere [10]. Acetylated aliquots of all TLEs were additionally measured using a GC-FID system (Fisons Instruments GC 8000 series (Fisons Instruments, Ipswich, United Kingdom), Fisons Instruments HRGC MEGA 2 series (Fisons Instruments, Ipswich, United Kingdom), and Thermo Scientific Trace 1300 Series (Themo Fisher Scientific, Waltham, MA, USA) to monitor the performance of the HPLC-APCI-MS system [10]. The response factor between the 5-α-cholestane and the DAGE C_18:18_ standard was 1.6:1 on the GC-FID. Specific lipid production rates and product-to-product yields were determined using the DAGE C_18:18_ and C_46_ GDGT. However, due to an application error of the DAGE C_18:18_ in some samples, the 5-α-cholestane and C_46_ GDGT were used instead in these cases. The response factor between the 5-α-cholestane and DAGE C_18:18_ was considered in the calculations. Appendix A indicate which samples were quantified with the 5-α-cholestane.

## 3. Results

### 3.1. Specific Total Lipid Production Rates Depend on Culture Conditions

Specific production rates (µmol g^−1^ h^−1^) were determined for each lipid separately and for total lipids at all culture conditions (Appendix A). For all the experiments performed, the mean values of the total lipid production rates lay between 0.015 and 0.070 µmol g^−1^ h^−1^, depending on the experimental setting (Figure 3). Mean values were calculated for each of the 48 experimental settings (including different incubation times): 36 in the 117 mL, and 12 different experimental settings in the 570 mL bottles.

Even though the environmental conditions for the 117 mL serum bottles were the same for two sets of four replicates each, the slightly different total incubation time and the varying incubation periods had an obvious influence on the CH_4_ production (MER; see Appendix A) and lipid production rates of *M. marburgensis.* Moreover, the MER was higher at higher atmospheric pressures and at lower liquid volumes, and the total lipid production rate tended to be higher at shorter incubation times for the V/V = const. settings. No clear trend was observed for the n/n = const. settings (Figure 3), except the significantly higher MER values for the experiments performed with small liquid volumes (see Appendix A).

On average, the specific total lipid production rates were higher at V/V = const. than at n/n = const. Apart from these observations, there were no consistent patterns observed in the 117 mL bottles. Neither headspace pressure nor varying the volume of liquid medium clearly increased or decreased the specific total lipid production rates.

### 3.2. Product-to-Product Yield Followed the Trends of Specific Lipid Production Rates

The product-to-product yield is given as µmol lipid C-mol^−1^ biomass for each lipid separately and for total lipids at all culture conditions (C-mol^−1^ depicts per mole of carbon). On average, the total lipid yield lay between 50 and 160 µmol C-mol^−1^ (Figure 4). Overall, the lipid yields followed the same patterns as the specific lipid production rates. Especially, different incubation times significantly changed the total lipid yield. The total lipid yields at V/V = const. in the 117 mL bottles tended to be high at 1.5 bar/80.25 h, exceeding 110 µmol C-mol^−1^. In contrast, in the n/n = const. experiments, the average total lipid yields in the 117 mL bottles were always below 110 µmol C-mol^−1^. In the 570 mL bottle V/V = const. experiments, all samples showed values around 60 µmol C-mol^−1^. However, like the specific lipid production rates, experiments at n/n = const. in the 570 mL bottles showed an increase in total lipid yield from smaller to larger volumes (59 ± 37 µmol C-mol^−1^ at smaller volumes to 118 ± 41 µmol C-mol^−1^ at larger volumes at 1.5 bar; Figure 4).

The total lipid production rates and yields at V/V = const./1.1 bar were very similar in the 117 mL bottles grown for 93.12 h to those grown in 570 mL bottles for 70.72 h (dark blue bars at V/V = const. in Figure 3 and Figure 4). The experiments conducted at 1.5 bar did not result in such similar outcomes.

### 3.3. High Variability of Lipid Ratios Challenges Maintenance of Constant Lipid Quality

The focus of this study was to investigate the quality and product ratio of the produced core lipids. We found that archaeol and the tetraether lipids together made up more than 99% of total lipids in *M. marburgensis*. The proportion of the GDDs in this study was, in general, below 0.2%, but it reached about 0.5% of total lipids at large volumes at V/V = const./1.5 bar/80.25 h and about 0.3% at large volumes at n/n = const./2.0 bar/102.65 h in the 117 mL bottles. The relative amount of archaeol varied between 20% (570 mL bottles, large volumes at n/n = const./1.1 bar/75.25 h) and almost 80% (117 mL bottles, small volumes at n/n = const./1.1 bar/102.65 h). On average, *M. marburgensis* produced about 50% tetraether lipids, depending on the culture conditions (Figure 5). At V/V = const., tetraether lipids comprised 39 ± 4%, whereas at n/n = const., they made up 59 ± 8% in the 117 mL bottles. In the 570 mL bottles, they constituted 49% at V/V = const. and even 65% at n/n = const. Thus, the condition n/n = const. overall led to a higher proportion of tetraethers vs. archaeol compared to V/V = const., where archaeol was the most abundant membrane lipid. However, this ratio relied on pressure and incubation time as well, as the high proportions of archaeol in the smaller volumes at n/n = const./1.1 bar clearly demonstrate (Figure 5).

The proportions of the different groups of tetraethers—GDGTs, GMGTs, and GTGT-0a—did not vary largely (Figure 6). The GDGTs were the most abundant tetraethers at all culture conditions, accounting for about 80 to 95% of total tetraether lipids. The GMGTs were the second most abundant group of tetraethers, with an average 4 to 18% of total tetraethers. The relative amount of GTGT-0a lay well below 1% of total tetraethers at most culture conditions. It approached the 1% limit in all samples cultured at V/V = const./93.12 h in the 117 mL bottles, and it even reached more than 2% in some of the smaller volume replicates at n/n = const./1.1 bar in the 117 mL bottles at 102.65 h and in the 570 mL bottles at 75.25 h. However, the standard deviations for GTGT-0a in all cases are very high; therefore, the fluctuations of GTGT-0a are not robust, and the results should be treated with care.

Compared to GDGT-0a and -0b, the relative amounts of GDGT-0c are negligible (below 1% of total GDGTs; Figure 7). With an average 70 to 90%, GDGT-0a shows the highest relative amount among the GDGTs. The highest average proportion of GDGT-0a (91.5%) was measured at large volumes for V/V = const./1.1 bar/93.12 h in the 117 mL bottles. In contrast, the lowest average proportion of GDGT-0a (70.6%) was measured at large volumes for n/n = const./1.5 bar/75.25 h in the 570 mL bottles. The proportion of GDGT-0b and -0c relative to GDGT-0a (degree of methylation) in the 117 mL bottles at V/V = const. (especially those at 93.12 h) was, on average, lower than in the 117 mL bottles at n/n = const., and in all the experiments in the 570 mL bottles.

The GMGTs did not show a predominance of the 0a isomers, as shown for the GDGTs (Figure 8). In contrast to GDGT-0c, GMGTs-0c made up between 1 and 7% of total GMGTs at most growth conditions. However, in the 117 mL bottles at V/V = const./93.12 h, GMGTs-0c accounted for less than 1% of total GMGTs, which did not occur in any of the other sets of culture conditions. The relative proportion of GMGTs-0c was highest in the 117 mL bottles at n/n = const./82.82 h in the medium volume at 2.0 bar and in the 570 mL bottles at n/n = const. (except the small volume at 1.1 bar). There, it even reached 8% of total GMGTs. Depending on the culture condition, either GMGTs-0a or GMGTs-0b were predominating, whereas GMGTs-0a were dominant more often. The relative proportion of GMGTs-0b varied between 20% and almost 60% of total GMGTs. The ratio (1 × GMGT-0b + 2 × GMGT-0c)/(GMGT-0a + GMGT-0b + GMGT-0c), indicating the degree of methylation, tended to be higher at conditions with V/V = const., but only in the 117 mL bottles. For the 570 mL bottles, a higher degree of methylation was only observed for the cultures at V/V = const./1.5 bar (Figure 9).

### 3.4. Impact of Interaction-Area-to-Volume Ratio on Growth

As expected, the experiments performed with a small liquid volume, i.e., a high gaseous volume and, therefore, a high number of gaseous substrates, showed the highest end point optical density (OD_end_, Figure 10). While for experiments with n/n = const. a longer incubation time led to a higher OD_end_, this was not observed for the V/V = const. settings. Further, at V/V = const., a higher pressure resulted in a higher OD_end_, while for n/n = const., the OD_end_ decreased with increasing pressure. However, this trend for n/n = const. could be reversed when dealing with larger volumes, as the 82.82 h pressure series at large volumes may imply. A very remarkable result was the extremely high OD_end_ value accomplished for the n/n = const. experiments at small volumes and 1.1 bar.

## 4. Discussion

In this study, *M. marburgensis* was cultivated under different cultivation conditions and different scales. One of the most remarkable findings was the strong influence of the total incubation times and gassing intervals, not only on the specific lipid production rates and product-to-product yields, but also on the lipid ratios. For instance, in the 117 mL bottles in the small volumes at n/n = const./1.1 bar, archaeol production rates were much higher at a total incubation time of 102.65 h compared to a total incubation time of 82.82 h (Figure 5). We cannot explain this discrepancy yet; however, the pressure data showed a strikingly different growth pattern between these two settings within the first 40 h of incubation. Nevertheless, this is only the most extreme example of a series of cases within this study, where incubation times and intervals made the difference. Another important observation of this study was that the total lipid production rates and product-to-product yields in the 117 mL bottles were higher at V/V = const. than at n/n = const., on average. The reason for that pattern is yet unknown. However, it is noteworthy that the lipid production rates and yields are, in general, more similar between the 117 mL and 570 mL bottles in the case of n/n = const. compared to V/V = const. (with some exceptions). Our data clearly stress that the incubation times and intervals need to be strictly equal when attempting to scale lipid production with *M. marburgensis*. From the current data, up-scaling at n/n = const. seems more promising, not only based on quantitative considerations. Various lipid ratios at V/V = const. in the 117 mL bottles varied in other experimental settings. Generally, lipids that are considered as more specific to *M. marburgensis*, such as GMGTs, extra-methylated GDGTs, and GMGTs [14], were less abundant at V/V = const. in the 117 mL bottles, respectively. In contrast, more prominent membrane lipids, such as archaeol and GDGT-0a (e.g., [4,50]), were more abundant at the other conditions. This observation supports a scale-up at n/n = const., especially when a greater variety and higher yield of the minor lipids could be harvested.

An unexpected, yet intriguing, finding was the extraordinarily high growth (and OD_end_) in the smaller volumes at n/n = const. at 1.1 bar, when the liquid volume-to-headspace-ratio was by far the smallest applied (Figure 9). The explanation for this finding could be the very high ratio of interaction area vs. total liquid volume, in which the organisms can grow. This finding indicates that the scale-up conditions must be well-defined to consider the specific gas transfer coefficient (k_L_a value) or the gas transfer rate to be able to align the specific growth rate and/or cell concentration. Moreover, it would be interesting to test even higher ratios of interaction area vs. liquid volume in future studies. If the optimal ratio is found, it could then be used for further scale-up settings for industrial purposes. The settings showing the highest OD_end_ are n/n = const. at small volumes and 1.1 bar at 117 mL (102.65 h) and 570 mL (75.25 h). These settings also reveal a relatively high amount of GDGT-0a and GMGTs-0a compared to the other experiments in the n/n = const. settings.

The results shown here indicate the need to examine the lipid production rates and yields and the composition of the lipid inventory under different cultivation conditions in closed batch cultures with the goal of identifying the scaling parameters for a reproducible archaeal-lipid-production pipeline. This is necessary because now the drawbacks and issues are known to successfully start the scale-up of archaea lipid production for the mass utilization of methanogens in Archaea Biotechnology. However, it must be noted that the closed batch growth of methanogens differs from the fed-batch or continuous culture growth of methanogens. Such differences in the cultivation set-up could induce even another lipid production characteristic. Other advantages of employing methanogens for archaeal lipid production are the ability to excrete proteinogenic amino acids into the growth medium [51] and to produce CH_4_ [19] in addition to ether-based lipids [10]. This makes it worthwhile to strongly consider *Methanothermobacter* spp. and other methanogens as chassis to produce various value-added products in biotechnology alongside their use as CH_4_ cell factories. Once such an integrated biotechnological production platform is established, hydrogenotrophic, autotrophic methanogens could replace the synthetic production of ether lipids, which is based on non-renewable resources. Hence, a methanogen-based lipid-production bioprocess could make use of the H_2_ generated from excess renewable energy production, such as wind or solar power, or from biohydrogen production, and the CO_2_ from renewable sources, e.g., from bioethanol production. A utilization of methanogens as archaeal cell factories in biotechnology and in biorefinery concepts seems already reasonable.

## 5. Conclusions

Varying the environmental conditions and the incubation periods has a significant impact on the growth, the MER, and the lipid production rate of *M. marburgensis*. This study shows that keeping the particle number constant (n/n = const.) at different pressure settings leads to a higher variability in the lipid pattern than keeping a constant ratio between the liquid and gaseous volumes (V/V = const). Besides this new insight, the study shows the significant influence of different incubation periods for the same environmental setting. For biotechnological and industrial purposes, the most important outcome of this study is the potential for optimizing the process by finding the right ratio between the interaction area and the total liquid volume. Here, future studies must be performed to optimize current approaches.

## Figures and Tables

**Figure 1 bioengineering-09-00169-f001:**
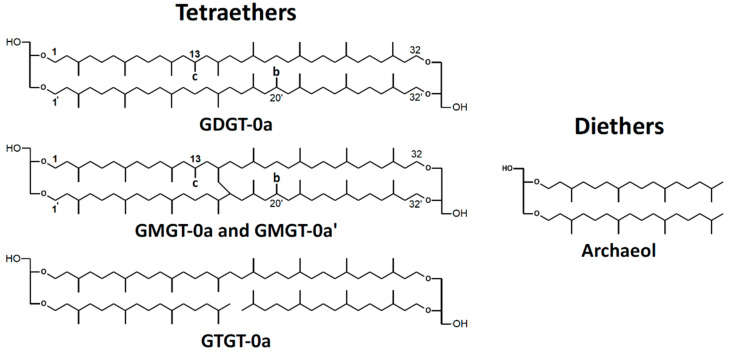
Structures of polar headgroups and core lipids found in *M. marburgensis*. Abbreviations of core lipids are: GDGT (glycerol dialkyl glycerol tetraether), GMGT (glycerol monoalkyl glycerol tetraether), and GTGT (glycerol trialkyl glycerol tetraether). “0”: zero rings in the alkyl chains; “a”, “b”, and “c”: no, one, or two additional methyl groups, as indicated in the structures [14]. Note that the exact positions of the additional methyl groups and the covalent carbon–carbon bonds between the isoprenoid chains of GMGT-0 are unknown.

**Figure 2 bioengineering-09-00169-f002:**
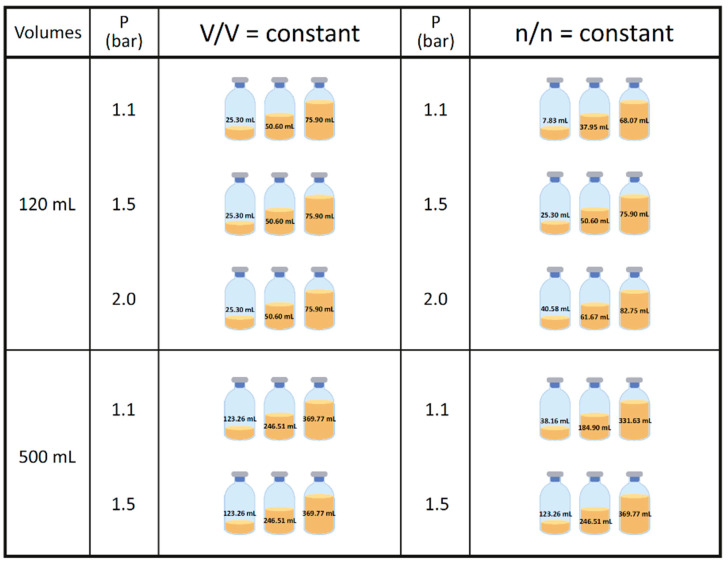
Schematic overview of the experimental set-up in 117 mL serum bottles. Pressure is given in bar relative to atmospheric pressure. Experiments in the 117 mL bottles were performed in quadruplicates (n = 4), and samples obtained after two different incubation intervals and repeated twice (N = 2). The final media and inoculum volumes for each experiment can be found in Appendix A.

**Figure 3 bioengineering-09-00169-f003:**
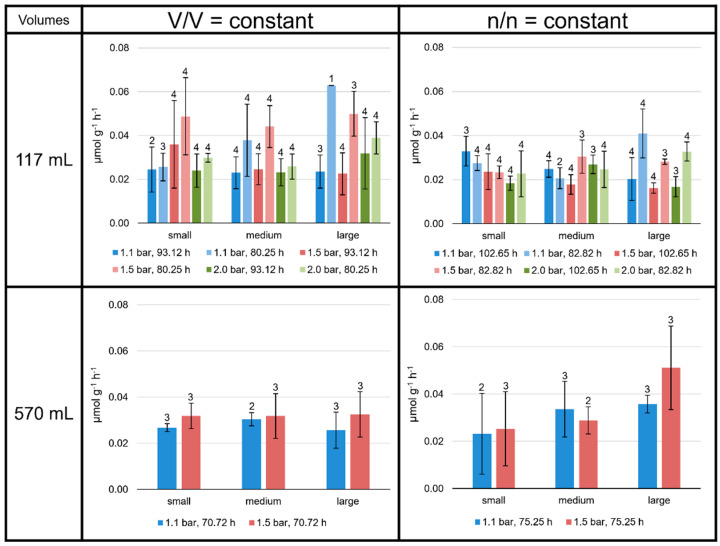
Specific total lipid production rates in *M. marburgensis* cultures. Pressure is given in bar relative to atmospheric pressure. Production rates are given in µmol g^−1^ h^−1^. Errors are standard deviations. Numbers above bars indicate number of samples. Samples quantified with 5-α-cholestane standard are indicated in Appendix A.

**Figure 4 bioengineering-09-00169-f004:**
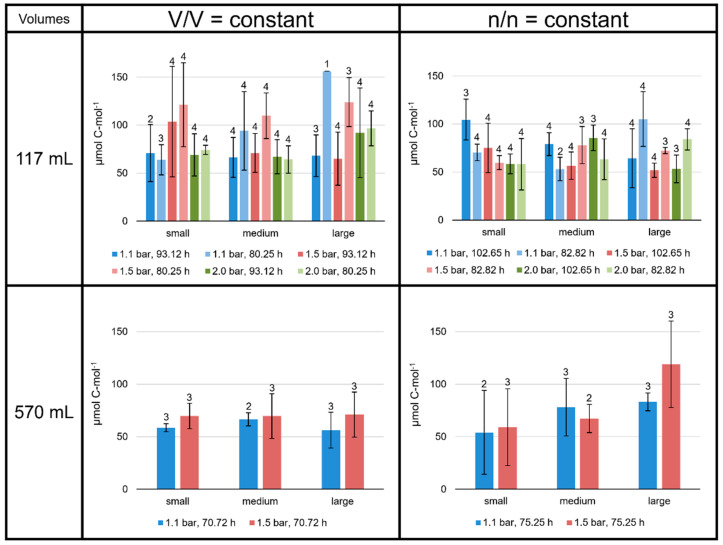
Product-to-product yields of total lipids in *M. marburgensis* cultures. Pressure is given in bar relative to atmospheric pressure. Yields are given in µmol C-mol^−1^. Errors are standard deviations. Numbers above bars indicate number of samples. Samples quantified with 5-α-cholestane standard are indicated in Appendix A.

**Figure 5 bioengineering-09-00169-f005:**
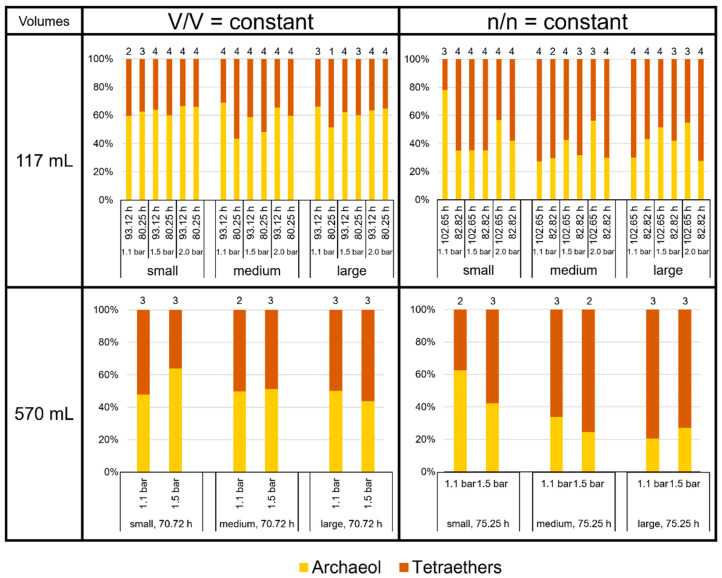
Lipid ratios of archaeol to tetraethers in *M. marburgensis* cultures in %. Pressure is given in bar relative to atmospheric pressure. Numbers above bars indicate number of samples. Samples quantified with 5-α-cholestane standard are indicated in Appendix A.

**Figure 6 bioengineering-09-00169-f006:**
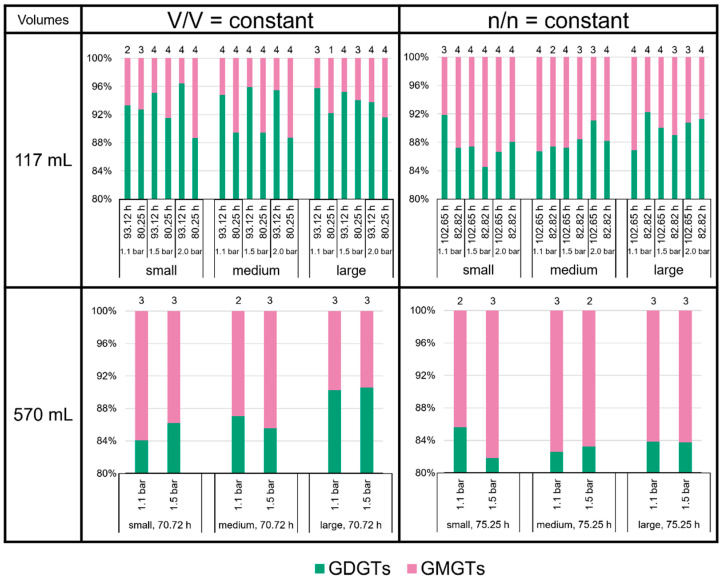
Tetraether lipid (GDGTs and GMGTs) ratios in *M. marburgensis* cultures in %. Pressure is given in bar relative to atmospheric pressure. Numbers above bars indicate number of samples. Samples quantified with 5-α-cholestane standard are indicated in Appendix A.

**Figure 7 bioengineering-09-00169-f007:**
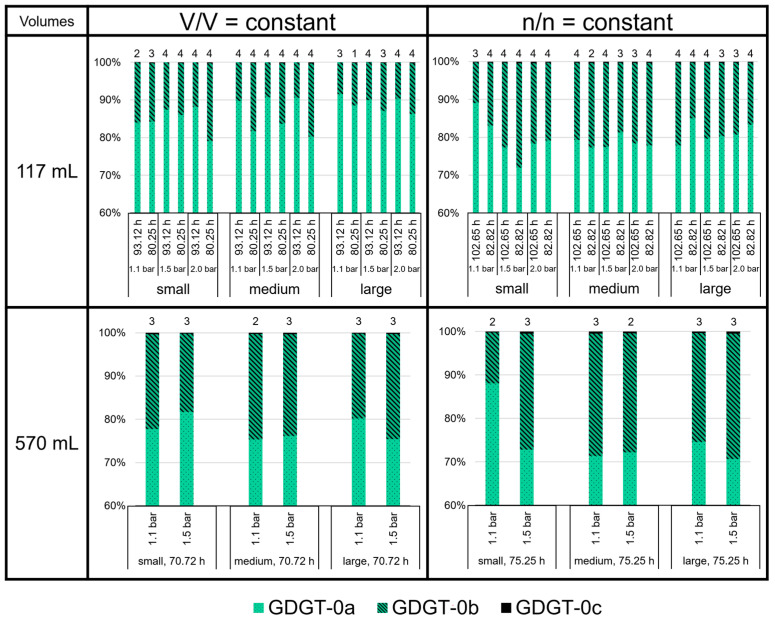
GDGT lipid (GDGT-0a, GDGT-0b, and GDGT-0c) ratios in *M. marburgensis* cultures. Pressure is given in bar relative to atmospheric pressure. Numbers above bars indicate number of samples. Samples quantified with 5-α-cholestane standard are indicated in Appendix A.

**Figure 8 bioengineering-09-00169-f008:**
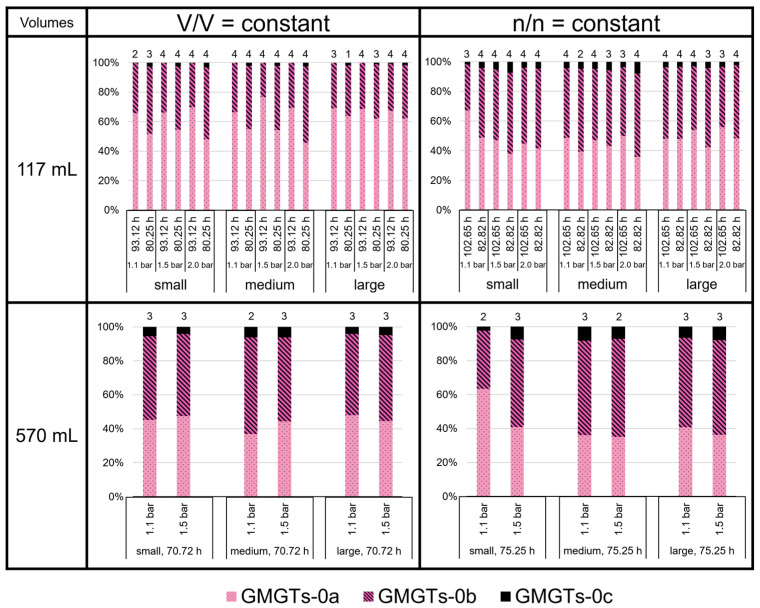
GMGT lipid ratios (two isomers each of GMGT-0a, GMGT-0b, and GMGT-0c) in *M. marburgensis* cultures. Pressure is given in bar relative to atmospheric pressure. Numbers above bars indicate number of samples. Samples quantified with 5-α-cholestane standard are indicated in Appendix A.

**Figure 9 bioengineering-09-00169-f009:**
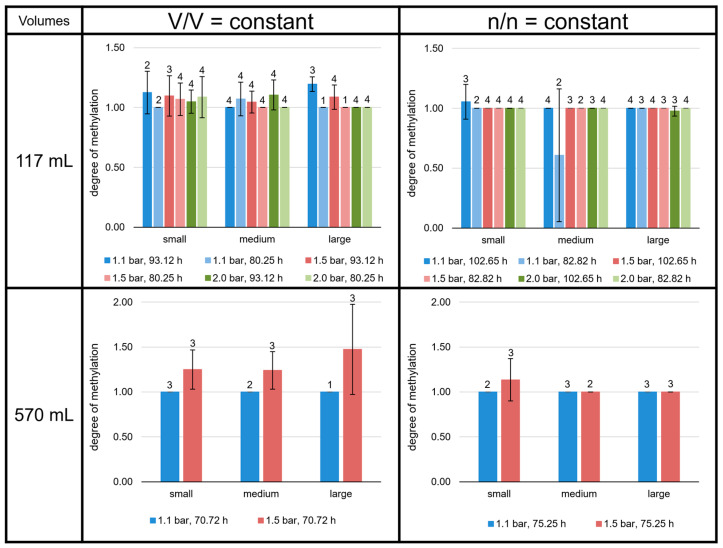
Degree of methylation of the *M. marburgensis* cultures. Pressure is given in bar relative to atmospheric pressure. Errors are standard deviations. Numbers above bars indicate number of samples.

**Figure 10 bioengineering-09-00169-f010:**
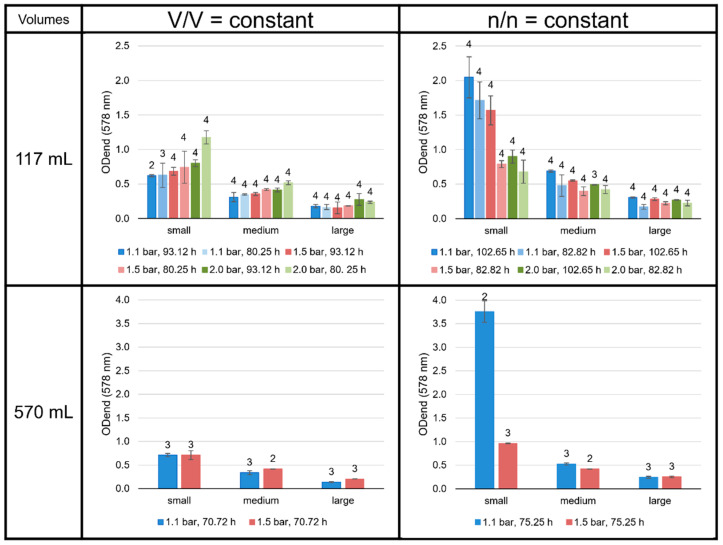
OD_end_ (578 nm) of the *M. marburgensis* cultures. Pressure is given in bar relative to atmospheric pressure. Errors are standard deviations. Numbers above bars indicate number of samples.

## Data Availability

The datasets used and/or analyzed during the current study are available from the corresponding author on reasonable request.

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
