# Peer review of "Quantitative Analysis of Core Lipid Production in Methanothermobacter marburgensis at Different Scales"

_bioengineering, 2022, doi:10.3390/bioengineering9040169_

Round 1

Reviewer 1 Report

Dear Authors

I recommend your revised manuscript for publication in Bioengineering journal.

One suggestion, line 55: Archaea – use a capital letter.

Reviewer 2 Report

The changes made in response to reviewer comments are adequate

Reviewer 3 Report

Review Bioengineering-1594247 Baumann et al 2022

Quantitative analysis of core lipid production in Methanothermobacter marburgensis at different scales

I want to thank the authors for the thorough revision of their manuscript, which is in my opinion ready for publication (after text/spell checking). It is a really interesting study that will certainly further the use of Archaea as (lipid) biotechnological platforms. I overall enjoyed reading the manuscript and interacting with the authors, which I hope was beneficial for both parts.

Minor comments/answers:

Lines 249-252: No value nor error are presented here, and the sentence is thus very vague.

Thank you for that comment. Again, we think that mean values and errors would not make sense in that context. Here, we just refer to the visual impression that the bars in the V/V=const graph tend to be higher on average than those of the n/n=const graph.

I understand and agree with the authors’ point. This might however be upsetting/disturbing for some readers and one must hence make sure that it is clearly stated that trends are drawn from visual impression. This also applies to my previous comment on Line 297.

This manuscript is a resubmission of an earlier submission. The following is a list of the peer review reports and author responses from that submission.

Round 1

Reviewer 1 Report

Dear Authors,

Your manuscript ID: bioengineering-1594247 contributes to the recognition of the possibility of application methanogens into other than widely known purpose therefore might be interested for other scientists. I have only few suggestions:

Lines 21, 34, 35 and whole manuscript,

Use a capital letter when write about microbial domain (Archaea).

Line 22

‘....produce methane from carbon dioxide or organic substrates.’ Be accurate, not only from carbon dioxide but carbon dioxide and hydrogen.

Line 165

2.2. Cultivation Medium

Some kind of system for flushing were used?

Lines 302

Figures 5-8, the captions and scale on the axes are barely visible.

Author Response

Reviewer #1:

Your manuscript ID: bioengineering-1594247 contributes to the recognition of the possibility of application methanogens into other than widely known purpose therefore might be interested for other scientists.

Thank you for this summary.

Lines 21, 34, 35 and whole manuscript, use a capital letter when write about microbial domain (Archaea).

Thank you. We changed to capital letter A where needed.

Line 22: “....produce methane from carbon dioxide or organic substrates.” Be accurate, not only from carbon dioxide but carbon dioxide and hydrogen.

Thank you. We edited the respective sentence.

Line 165ff: 2.2. Cultivation Medium – Some kind of system for flushing were used?

The flushing system and procedure is described in detail in Taubner & Rittmann, 2016. This reference has been cited in the manuscript where needed.

Lines 302: Figures 5-8, the captions and scale on the axes are barely visible.

Thank you. We increased the labelling of the axes.

Reviewer 2 Report

I have completed my review. It appears the portal has closed. You can find this below.
My recommendation is _Major Revision required_.

This manuscript by Baumann et al. investigates the impact of various growth conditions on the production of methane and archaeal lipids by /Methanothermobacter marburgensis. /The growth conditions are well described and standard metrics are reported. There are a number of minor writing and grammatical issues with the paper as described below. The information provided is a good starting point for optimizing conditions for scaling up growth. A short coming of the paper is that the authors do not take their own advice and use what was learned here to test optimized growth conditions. For example see this section below.

Lines 466-469: “The most important outcome of this study is the potential of optimizing the process by finding the right ratio between the interaction area and the total liquid volume. Here, future studies have to be performed to optimize current approaches.”

Minor comments:

Line 19: “Archaeal lipids do have a…” should be just “Archaeal lipids have…” Most likely an issue arrised from authors being from Germany/Austria

Line 34 The sentence is awkward. Suggest a change to “ Membrane lipids of archaea can be considered among the most outstanding adaptations of life.

Line 48 Should be rewritten. Use of “shortly” does not make sense.

Line 72-77: Long run-on sentence

Lines 78-99: The explanation of the core lipids and their nomenclature is presented best in the figure but the text is rather confusing. As is, it reads like they started to explain the nomenclature, then stopped and put the rest in the figure legend. This seems to need revision.

Figure 2. The volumes labeled on the flasks are hard to read. Please make the font larger.

Lines 204-207: The authors say that the cell pellets were stored at -20C until further analysis but the next step is lyophilization and not an analysis step.

Line 235:  “Of all” should be changed to “For all”.

Figures 3-8:  Font size should be increased to improve readability. For example, the bars and font in the top two panels of Fig. 9 look better.

Lines 300-301: Figure 5 – There are replicates but no error bars given. This visually would be nice and could strengthen conclusions

Lines 341-344: Figure 6 – Same comments as figure 5

Same as figures 7 and 8

Lines 406-438:  This text belongs in the introduction rather than the Discussion since it does not relate to the experiments/results of this paper.

Lines 455-456:  What is meant by “excess power”? 

Author Response

Reviewer #2:

This manuscript by Baumann et al. investigates the impact of various growth conditions on the production of methane and archaeal lipids by Methanothermobacter marburgensis. The growth conditions are well described and standard metrics are reported. There are a number of minor writing and grammatical issues with the paper as described below. The information provided is a good starting point for optimizing conditions for scaling up growth. A short coming of the paper is that the authors do not take their own advice and use what was learned here to test optimized growth conditions. For example, see this section below.

Lines 466-469: “The most important outcome of this study is the potential of optimizing the process by finding the right ratio between the interaction area and the total liquid volume. Here, future studies have to be performed to optimize current approaches.”

Thank you for this summary. Regarding your comment on taking our own advice: We totally agree with you that testing the optimized growth conditions would be the next step. However, optimization would definitely be the scope of a standalone study, and, as it is not part of this study, it shall be included in future ones.

Minor comments:

Line 19: “Archaeal lipids do have a…” should be just “Archaeal lipids have…” Most likely an issue arrised from authors being from Germany/Austria.

Thank you. We edited the phrase according to your suggestion.

Line 34 The sentence is awkward. Suggest a change to “ Membrane lipids of archaea can be considered among the most outstanding adaptations of life.

Thank you. We edited the phrase according to your suggestion.

Line 48 Should be rewritten. Use of “shortly” does not make sense.

Thank you. We exchanged “shortly” with “also referred to as”.

Line 72-77: Long run-on sentence

Thank you. We divided the sentence into two separate ones.

Lines 78-99: The explanation of the core lipids and their nomenclature is presented best in the figure but the text is rather confusing. As is, it reads like they started to explain the nomenclature, then stopped and put the rest in the figure legend. This seems to need revision.

Thank you. We agree to shorten this paragraph. We only left the most important points and the reference for the nomenclature introduced by Knappy et al. (2015).

Figure 2. The volumes labeled on the flasks are hard to read. Please make the font larger.

Thank you. Due to visualization issues, we would not like to increase the font size in the figure, as the labelling would stretch over the bottle, or rotate. However, the numbers can be easily seen in Table S1. We added this information to the figure’s caption.

Lines 204-207: The authors say that the cell pellets were stored at -20C until further analysis but the next step is lyophilization and not an analysis step.

Thank you. We agree, but the lyophilization is the first step of sample preparation to make them suitable for the analysis. Therefore, the lyophilisation is part of the analysis, more precise the preparation.

Line 235:  “Of all” should be changed to “For all”.

Thank you. We edited the phrase according to your suggestion.

Figures 3-8:  Font size should be increased to improve readability. For example, the bars and font in the top two panels of Fig. 9 look better.

Thank you. We increased the labelling of all mentioned figures.

Lines 300-301: Figure 5 – There are replicates but no error bars given. This visually would be nice and could strengthen conclusions

Thank you for the comment. Due to the vizualization of results on a scale from 0-100% it is not possible to add error bars. As we provide error bars in all other figures and given that we provide all data in the suppelmentary material, we kindly ask that you kindly accept our vizalization mode.

Lines 341-344: Figure 6 – Same comments as figure 5. Same as figures 7 and 8

Thank you. Same reply as above.

Lines 406-438:  This text belongs in the introduction rather than the Discussion since it does not relate to the experiments/results of this paper.

Thank you. The text was moved into the introduction.

Lines 455-456:  What is meant by “excess power”? 

Thank you. We provide examples for excess energy.

Reviewer 3 Report

Quantitative analysis of core lipid production in Methanothermobacter marburgensis at different scales

This manuscripts reports the qualitative and quantitative changes in the core lipid pool of a model methanogenic archaeon, Methanothermobacter marburgensis, under controlled conditions of substrates load, gas/medium volumes and hydrostatic pressure as 1) a proof-of-concept that M. marburgensis can serve as a platform for the production of particular archaeal lipids and 2) a first step towards upscaling such lipid production.

As an archaeal lipidomist, I can confirm the field’s interest for such studies, which help constrain the parameters controlling lipid production in Archaea and push forward their use as industrial platforms. I have enjoyed reading the manuscript, which contains some interesting results. I however had a really hard time navigating it for two reasons: 1) the actual results are sometimes hard to find and not well presented and/or explained, which in my opinion is also the case for most of the figures (some, like Fig. 7, are really hard to read), 2) some parts of the manuscript are really obscure and ill-organized. I thus do not think the manuscript can be published as it stands.

I listed below some comments that might help improve the overall readability of the manuscript and should be addressed for publication.

General comments:

Introduction: In my opinion, the introduction as it stands does not fulfil its functions, i.e., clearly stating the aim and the hypothesis of the manuscript, what methodology and why this particular one has been employed, and what literature data it has been built upon. Two major drawbacks are the unclear/unconvincing evidence that M. marburgensis is a good model for archaeal lipid production and that recording core lipids instead of intact polar lipids variations is the proper methodology.

Results: Most of the actual data remain unfortunately completely untapped, resulting in a very vague results section where it was hard to 1) find the important data and 2) compare between experiments/lipid structures. For instance, Tables S4 to S9, which are huge, are referred to only a few times without stating the exact data the reader should pay attention to (no values are given!) and the interesting data they contain is thus not properly used to support the conclusions of the manuscript. Errors are most often not stated in the text while they seem pretty large on the figures (e.g., Fig. 3). Figures 4 to 8 are pretty hard to read because too small and hard to distinguish color code.

Discussion: As for the Results section, the Discussion section remains pretty vague and the conclusions/hypotheses are often not well supported by the data, or at least those actually presented in the manuscript. While I agree there is some really interesting data in the manuscript, they need to be more properly introduced/described to reveal their potential. As a side note, I found the measurements at different pressures very interesting since they are only a limited number of such studies in the literature (which are not mentioned/compared to), but those are unfortunately not discussed at all in the current manuscript. I think they could also be a valuable add-on for later versions of the manuscript.

Minor comments:

Line 37: I do not think that “anoxic environments” can be considered extreme environments.

Line 39-40: If eukaryotic and bacterial lipid side chains are acyl chains, then archaeal ones are isoprenoid chains, and not alcohols. On the other hand, if you say archaeal side chains are isoprenoid alcohol, then eukaryotic and bacterial side chains should be fatty acids. I do not mind either of these, but I would rather go for acyl vs. isoprenoid chains.

Line 41: Ether-bound instead of ether-bond. Also, stability is a very general term which is almost senseless on its own. Maybe add thermal/chemical/enzymatic stability.

Lines 46-47: Why is that? For instance, tetraether lipids are harder to handle than their bilayer-forming C20 counterparts, which still possess all the interesting characteristics of archaeal lipids.

Lines 47-48: I would appreciate a better description of the distribution of tetraether lipids in Archaea, which would direct the reader towards other archaea that might be suitable for lipid production. For instance, thermoacidophilic archaea like Sulfolobus acidocaldarius have higher proportions of tetraether lipids than most methanogens, are easier to grow (they are fast-growing aerobes) and are already industrial model organisms (S. acidocaldarius can easily be grown in > 300 l fermenters, has a diverse genetic tool box, etc.). As mentioned above and below, I am not convinced, as it stands, that methanogens and M. marburgensis would be suitable industrial platforms for lipid production (although they are for a whole lot of other biotechnologies, like carbon-related applications mentioned in the manuscript).

Lines 57-60: These are also produced by other archaea, sometimes even by a single species (e.g., Ignisphaera aggregans (Knappy et al., 2011), Cuniculiplasma divulgatum (Golyshina et al., 2016) and Pyrococcus furiosus (Tourte et al., 2020)). Refer to my previous comment.

Lines 83-91: I would appreciate a better description of M. marburgensis core lipid composition (relative amounts of each compound) to compare with those described in the manuscript. An introduction of the term “core lipids” and how they differ from “intact polar lipids” would also help non-lipidomist readers. What are the head groups expected for M. marburgensis? And why not study the variations induced by the tested conditions on the IPL composition rather than that of the CL? From the biotechnological angle, I am not convinced that CL are the most prominent choice either since – I think – they do not form lipid bodies, e.g., films, liposomes, like their IPL counterparts or give them particular physicochemical properties, e.g., enhanced fusion rates, higher permeability.

Lines 100-101: Examples of similar studies in M. marburgensis or other methanogens would be appreciated as they would indicate how M. marburgensis lipid composition is expected to vary with the tested parameters.

Lines 108-109: Why would one expect these particular parameters to impact M. marburgensis’ lipid composition? Different hydrostatic pressures were also applied but the resulting variations are completely omitted here and in the rest of the manuscript.

Figure 1: Why mention GMD and GDD if they are is such low amounts and not considered further in the manuscript (and might be degradation products of GMGT and GDGT, respectively)? Carbon numbers are too small to read and the difference between b and c is not clear (here it seems b corresponds to a methylation in the indicated position and c in the other, whereas b and c are actually with one and two methylations).

Lines 116-120: Genetic background can have drastic effects on lipid compositions. Are these strains strictly the same? How do they differ genetic and lipidomic wise?

Figure 2: Lipid production in small volumes was assessed at 1.1, 1.5 and 2.0 bars, but only at 1.1 and 1.5 bars in large volumes. Any reason why 2.0 bars cannot be applied to the large volumes? I would have expected the glass bottles used to be able to withstand such overpressures even at 65 °C.

Line 156: I personally find the V/V and n/n = constant a bit unclear and even misleading. What does each letter stand for? Liquid vs. gas? For instance, since M. marburgensis consumes the gaseous substrates, the n/n is not constant. It would be more accurate in my opinion to say that the starting point is the same. Similarly in Table S1, the volumes indicated are those of the medium and inoculum while what really matters is that of the gaseous substrate (especially for calculating the amount of substance n). I also quickly recalculated these n in the different n/n = const. settings and, although I might have made some mistake, I do not find constant n (for instance, for the “small” settings, I found n = 4.9 and 5.6 mmol at 1.1 and 1.5 bars).

Line 177: Did you use cultures with the same cell concentrations/optical density? Was the number of cells the same at the beginning of each replicates?

Lines 211-213: Please indicate how much standard was used for each analysis. Why use two different extraction standards (cholestane and DAGE), and not only cholestane, especially considering the application error of DAGE in some samples?

Lines 218-219: Please introduce HPLC and APCI.

Line 236: Please provide standard deviations. How are the values distributed between 0.015 and 0.070 µmol g-1 h-1?

Lines 249-252: No value nor error are presented here, and the sentence is thus very vague.

Figure 3: This figure clearly shows that there are no real difference in specific total lipid production rates between growth conditions, especially considering the large error bars. Any explanation why they are this big? (and why that of V/V = constant 117 ml large 1.1 bar 80.25 h is so small compared to the others?)

Line 258: The way section 3.1 is presented suggests that there were no real trends between specific total lipid production rates and the tested parameters. Yet, the title of section 3.2 states that product-to-product yield followed the same trends than specific total lipid production rates, meaning there were no trend either? This is really unclear.

Lines 269-271: Instead of writing below 60 and almost 120 µmol C-mol-1, indicate actual values and errors.

Lines 292-293: 39 and 49 % are averaged proportions of tetraethers. Indicate the deviations across samples.

Line 297: Indicate the actual values for better comparison/understanding.

Line 314: Where can these data be found? Do GTGT0a specific production rates correlate with any of the tested parameters?

Lines 334-340: It appears the degree of methylation is calculated here as the ratios GMGT-0b/GMGT-0a or GMGT-0c/GMGT-0a while, following the ring index (averaged number of cycles per molecule) calculations described by Schouten and colleagues, it would be more accurate to calculate this as follows: (1 x GMGT-0b + 2 x GMGT-0c)/(GMGT-0a + GMGT-0b + GMGT-0c). Again, where can the values be found so that comparison could be made more easily. A plot of methylation degrees across conditions could also be interesting to visualize.

Lines 358-359: This seems true only in very specific settings, i.e., small volumes where it is not even true for all the experiments in small volumes (see for instance 1.1 bar, 102.65 and 82.82 h are equivalent). The same applies for lines 360-361: there are some examples of n/n constant cultures where higher pressures yielded higher ODend and vice versa for V/V constant.

Lines 383-385: What is this assumption based on? It is not that clear to me, especially considering the rather large error bars already present here with strictly controlled conditions.

Lines 385-386: Again, I do not understand why. Cultures at V/V constant often showed higher lipid production rates and product-to-product yields than cultures at n/n, and are thus, in my opinion, more promising in the prospect of upscaling cultures for lipid production.

Line 394: How does this compare to previous studies? Could this be even more increased, for instance by shaking the cultures?

Lines 434-438: Please reference this (for instance, Salvador-Castell et al. 2019, 10.3390/ijms20184434). This is also the first mention to squalene in the manuscript and I do not understand why it comes this late if it can be such a valuable compound. Does M. marburgensis produce squalene? Again, working with CL here probably degraded squalene and other such hydrocarbon derivatives preventing from understanding how the tested parameters control its production in M. marburgensis.

Author Response

Reviewer #3:

This manuscript reports the qualitative and quantitative changes in the core lipid pool of a model methanogenic archaeon, Methanothermobacter marburgensis, under controlled conditions of substrates load, gas/medium volumes and hydrostatic pressure as 1) a proof-of-concept that M. marburgensis can serve as a platform for the production of particular archaeal lipids and 2) a first step towards upscaling such lipid production.

Thank you for this summary.

As an archaeal lipidomist, I can confirm the field’s interest for such studies, which help constrain the parameters controlling lipid production in Archaea and push forward their use as industrial platforms. I have enjoyed reading the manuscript, which contains some interesting results. I however had a really hard time navigating it for two reasons: 1) the actual results are sometimes hard to find and not well presented and/or explained, which in my opinion is also the case for most of the figures (some, like Fig. 7, are really hard to read), 2) some parts of the manuscript are really obscure and ill-organized. I thus do not think the manuscript can be published as it stands.

Thank you for your comments. We are pleased that you enjoyed the topic of the manuscript. We hope that the changes that were introduced to the manuscript could improve the organisation and the possibility to understand the content of the manuscript. We sincelily hope that we improved the manuscripts according to your comments and we also hope that we could raise the figure and running text quality.

I listed below some comments that might help improve the overall readability of the manuscript and should be addressed for publication.

Thank you so much for the thorough review. We really appreciate that you did such a deep scientific analysis of our article! We hope that the modifications to the manuscript and the replies to your comment here make it possible to publish our manuscript.

General comments:

Introduction: In my opinion, the introduction as it stands does not fulfil its functions, i.e., clearly stating the aim and the hypothesis of the manuscript, what methodology and why this particular one has been employed, and what literature data it has been built upon. Two major drawbacks are the unclear/unconvincing evidence that M. marburgensis is a good model for archaeal lipid production and that recording core lipids instead of intact polar lipids variations is the proper methodology.

Thank you. We are aware of the fact that intact polar lipids may give us additional information (see Yoshinaga et al., 2015 for M. thermautotrophicum), especially under nutrient limitation and hydrogen depletion. In the study by Yoshinaga et al. (2015), they especially focussed on the variations of the headgroups, but did not take care for the composition of the core lipid composition. Since both M. thermautotrophicus and M. marburgensis are producing a collection of very diverse core lipids, we designed our experiments to specifically monitor changes of the core lipid inventory, which were our target compounds from the very beginning, since we were aiming for the maximum core lipid production under varying bioprocess/cultivation conditions due to biotechnological reasons. Moreover, we also aimed to collect unbiased compositions of core lipids. Depending on the extraction procedure, the composition of core lipids might be compromised (see Cario et al., 2015).

After having re-assessed the introduction we found that the introduction clearly states the goals and hypotheses and also introduces why M. marburgensis would be a well-chosen organism for core lipid production Archaea Biotechnology (please see also the comment further down here and in the introduction). M. marburgensis has a strong history in terms of publications regarding growth to high cell densities in bioreactors, biological methane production and biochemistry and was therefore explored with regard examine the lipid production across scales.

Results: Most of the actual data remain unfortunately completely untapped, resulting in a very vague results section where it was hard to 1) find the important data and 2) compare between experiments/lipid structures. For instance, Tables S4 to S9, which are huge, are referred to only a few times without stating the exact data the reader should pay attention to (no values are given!) and the interesting data they contain is thus not properly used to support the conclusions of the manuscript. Errors are most often not stated in the text while they seem pretty large on the figures (e.g., Fig. 3). Figures 4 to 8 are pretty hard to read because too small and hard to distinguish color code.

Thank you for the comment and improvement suggestions. We added some of the core findings of the Supplementary material to the main manuscript. However, we think that having the data packed in the Supplementary Tables helps to better compare the results than having lots of data presented in the text. The text should give an overview about the core results observed, while the supplementary tables give the reader the possibility to dive deep into the data.

With our study we attempted to reproduce (n = up to 4) and repeat (N = 2) all experiments. We attempted to examine gas conversion and core lipid production grown under conditions of H2/CO2 to CH4 conversion. Unfortunately, our strict experimental set-up did not result always in highly reproducible core lipid production patterns and rates under all investigated conditions. This is also a major outcome of this study, which is hard to scientists, but must be presented in an unbiased way. The reasons for this we attempted to discuss in very much detail in the discussion section. However, we do not hide our data and polish the results, but we clearly discussed possible solutions to this (partially) ambiguous outcome also in the frame of this study. We hope that this procedure can be valued by this reviewer.

Discussion: As for the Results section, the Discussion section remains pretty vague and the conclusions/hypotheses are often not well supported by the data, or at least those actually presented in the manuscript. While I agree there is some really interesting data in the manuscript, they need to be more properly introduced/described to reveal their potential. As a side note, I found the measurements at different pressures very interesting since they are only a limited number of such studies in the literature (which are not mentioned/compared to), but those are unfortunately not discussed at all in the current manuscript. I think they could also be a valuable add-on for later versions of the manuscript.

Thank you so much for you review. The partially ambiguous results that we report in our study are based on well discussed points that could have been arisen due to a varying the gas transfer rate, kLa, gas consumption/pressure reduction patterns, mixing regimes and associated growth rates. However, our study reports that various CL production patterns can be produced, and that the product-to-product yield and the rates can be clearly varied using our experimental set-up. However, we want to stress here again that we did n = up to 4 and N = 2 set-ups, which is quite uncommon in science – unfortunately. Hence, our study provides core insights into one of the major issues in cultivation experiments that are not very often attempted to be repeated, generally in science and in lipid production analysis, that is, that even under the same experimental set-up variability might occur.

Thank you also for the literature suggestions. We scanned them all in detail in the study by Cario et al. the scientists investigated, e.g. the effect of hydrostatic and not gas pressure. We also only used slight overpressure, which was the variable in the n/n and V/V experiments. We already cited Siliakus et al., 2017. In Oliver et al., 2020, A. fulgidus was grown at very high pressure, which is also much different from our growth conditions. We sincerely hope that we covered and cited most of the relevant scientific literature. We actually really take care about that.

Minor comments:

Line 37: I do not think that “anoxic environments” can be considered extreme environments.

Thank you. Sure, anoxic environments are not extreme, since they are very common, anyway they are still classified as ‘extreme’ with regard to any aerobic organism on our planet.

Line 39-40: If eukaryotic and bacterial lipid side chains are acyl chains, then archaeal ones are isoprenoid chains, and not alcohols. On the other hand, if you say archaeal side chains are isoprenoid alcohol, then eukaryotic and bacterial side chains should be fatty acids. I do not mind either of these, but I would rather go for acyl vs. isoprenoid chains.

Point taken and corrected accordingly.

Line 41: Ether-bound instead of ether-bond. Also, stability is a very general term which is almost senseless on its own. Maybe add thermal/chemical/enzymatic stability.

Point taken and corrected accordingly.

Lines 46-47: Why is that? For instance, tetraether lipids are harder to handle than their bilayer-forming C20 counterparts, which still possess all the interesting characteristics of archaeal lipids.

Thank you. This is a very good point. We focussed on tetraether lipids due to the fact that in nanobiotechology there is a high demand for tetraether lipids.

Lines 47-48: I would appreciate a better description of the distribution of tetraether lipids in Archaea, which would direct the reader towards other archaea that might be suitable for lipid production. For instance, thermoacidophilic archaea like Sulfolobus acidocaldarius have higher proportions of tetraether lipids than most methanogens, are easier to grow (they are fast-growing aerobes) and are already industrial model organisms (S. acidocaldarius can easily be grown in > 300 l fermenters, has a diverse genetic tool box, etc.). As mentioned above and below, I am not convinced, as it stands, that methanogens and M. marburgensis would be suitable industrial platforms for lipid production (although they are for a whole lot of other biotechnologies, like carbon-related applications mentioned in the manuscript).

Thank you for this comment. We modified this paragraph to make our reasoning clearer. Additionally, we have to make the point here that M. marburgensis is one of the most useful organisms for Archaea Biotechnology. The organism grows at a specific growth rate of 0.69 h‑1, which is higher than most Saccharolobales. M. marburgensis can be easily cultivated in bioreactors in chemostat culture using a cell retention system up to cell densities beyond 10 g L‑1 cell dry weight in bioreactors. Moreover, M. marburgensis can be grown in chemically defined minimal medium autotrophically at pH of 7 (the respective papers are cited in the introduction). Halophilic archaea and thermoacidophilic archaea require special bioreactors such as PEEK or special steel to withstand the high Cl-concentrations or the low pH due to SO42- production. In addition. the latter two archaeal groups are heterotrophic or mixotrophic (most of the Saccharolobales having lost the ability to fix CO2 due to a long domestication history). A complete overview on archaeal lipid production in archaea was done recently in Pfeifer et al 2022, Archaea Biotechnology, Biotechnology Advances. There we presented the state of the art with regard to lipid production and highlight the research done in this regard by halophilic, thermoacidophilic and autotrophic archaea.

Lines 57-60: These are also produced by other archaea, sometimes even by a single species (e.g., Ignisphaera aggregans (Knappy et al., 2011), Cuniculiplasma divulgatum (Golyshina et al., 2016) and Pyrococcus furiosus (Tourte et al., 2020)). Refer to my previous comment.

Thank you. We included these papers in our manuscript.

Lines 83-91: I would appreciate a better description of M. marburgensis core lipid composition (relative amounts of each compound) to compare with those described in the manuscript. An introduction of the term “core lipids” and how they differ from “intact polar lipids” would also help non-lipidomist readers. What are the head groups expected for M. marburgensis? And why not study the variations induced by the tested conditions on the IPL composition rather than that of the CL? From the biotechnological angle, I am not convinced that CL are the most prominent choice either since – I think – they do not form lipid bodies, e.g., films, liposomes, like their IPL counterparts or give them particular physicochemical properties, e.g., enhanced fusion rates, higher permeability.

We already stated above, why we decided to not measure intact polar lipids, but we agree that for future studies we should definitely include intact polar lipid measurements. Since we did not measure them for this study, we cannot include these data anymore. For that reason, we cannot provide any information on the polar head groups. We nevertheless hope that the manuscript is now suitable for publication.

Lines 100-101: Examples of similar studies in M. marburgensis or other methanogens would be appreciated as they would indicate how M. marburgensis lipid composition is expected to vary with the tested parameters.

Thank you. To our knowledge such studies do not exist. Your comment was actually one of the motivations to conduct this study. Our study would be the first one reporting this.

Lines 108-109: Why would one expect these particular parameters to impact M. marburgensis’ lipid composition? Different hydrostatic pressures were also applied but the resulting variations are completely omitted here and in the rest of the manuscript.

Thank you. We did not apply hydrostatic pressures but hyperbaric pressures. The goal was to assess if the particle number (n/n) at different scales or the gas to liquid volume ratio (V/V) at different scales alters the lipid production profile.

Figure 1: Why mention GMD and GDD if they are is such low amounts and not considered further in the manuscript (and might be degradation products of GMGT and GDGT, respectively)? Carbon numbers are too small to read and the difference between b and c is not clear (here it seems b corresponds to a methylation in the indicated position and c in the other, whereas b and c are actually with one and two methylations).

Thank you. We excluded GMD and GDD from both the text and Figure 1 and enlarged the carbon numbers.

Lines 116-120: Genetic background can have drastic effects on lipid compositions. Are these strains strictly the same? How do they differ genetic and lipidomic wise?

Thank you for the comment. The culture was initially obtained from the culture collection DSMZ GmbH (Braunschweig, Germany). We specified this in the manuscript.

Figure 2: Lipid production in small volumes was assessed at 1.1, 1.5 and 2.0 bars, but only at 1.1 and 1.5 bars in large volumes. Any reason why 2.0 bars cannot be applied to the large volumes? I would have expected the glass bottles used to be able to withstand such overpressures even at 65 °C.

Thank you. The used glass bottles can be used only to an overpressure of 1.5 bar, as recommended by the company.

Line 156: I personally find the V/V and n/n = constant a bit unclear and even misleading. What does each letter stand for? Liquid vs. gas? For instance, since M. marburgensis consumes the gaseous substrates, the n/n is not constant. It would be more accurate in my opinion to say that the starting point is the same. Similarly in Table S1, the volumes indicated are those of the medium and inoculum while what really matters is that of the gaseous substrate (especially for calculating the amount of substance n). I also quickly recalculated these n in the different n/n = const. settings and, although I might have made some mistake, I do not find constant n (for instance, for the “small” settings, I found n = 4.9 and 5.6 mmol at 1.1 and 1.5 bars).

Thank you. Yes, with V/V and n/n=const we always mean the starting point conditions, as of course this change over the time of the experiments. Regarding Table S1: We agreed on putting the volumes of the liquid medium and the inoculum into this table as these two parameters are important for preparing the experiments. As we stated the total volume of the two used kinds of bottles (117mL and 570 mL), the gaseous volume is easy to calculate from these numbers. We re-calculated the n/n=const settings and we received the same results as before. However, we identified the small mistake within your calculations: you have to use the absolute pressure values, not the relative ones (i.e., 2.1 and 2.5 bar instead of 1.1 and 1.5 bar). If doing so, this should result in the same values as the ones we calculated.

Line 177: Did you use cultures with the same cell concentrations/optical density? Was the number of cells the same at the beginning of each replicates?

Thank you. Our experiments were designed in a way that we did not disturb the suspension (cells) by sampling. Otherwise, our measurements would alter our experimental set-up.

The inoculum was prepared freshly each time under the same conditions and the cells were inoculated with the same procedure each time.

Lines 211-213: Please indicate how much standard was used for each analysis. Why use two different extraction standards (cholestane and DAGE), and not only cholestane, especially considering the application error of DAGE in some samples?

We added this information. We used two different extraction standards, because we quantified the contents of archaeol also on the GC-FID and always added two standards, just in case anything goes wrong with the upwork. As mentioned in the manuscript, we indeed had to use cholestane for some of the samples due to an application error. We hope that we could clarify this.

Lines 218-219: Please introduce HPLC and APCI.

Added.

Line 236: Please provide standard deviations. How are the values distributed between 0.015 and 0.070 µmol g-1 h-1?

Thank you for that comment. However, in our opinion, the determination of a standard deviation or a mean value doesn’t make sense in that case, as it is a multifactorial set-up. Here, we just give an overview about the min and max value over all different experiments for informational purposes.

Lines 249-252: No value nor error are presented here, and the sentence is thus very vague.

Thank you for that comment. Again, we think that mean values and errors would not make sense in that context. Here, we just refer to the visual impression that the bars in the V/V=const graph tend to be higher on average than those of the n/n=const graph.

Figure 3: This figure clearly shows that there are no real difference in specific total lipid production rates between growth conditions, especially considering the large error bars. Any explanation why they are this big? (and why that of V/V = constant 117 ml large 1.1 bar 80.25 h is so small compared to the others?)

Thank you. We agree that no clear trend can be seen in Figure 3, as also stated in section 3.1. The error bar of of V/V = constant 117 ml large 1.1 bar 80.25 h is so “small” (equals zero) because we just unfortunately had one single sample for the analysis (as indicated by the small numbers above the error bars) and therefore no error can be calculated.

Line 258: The way section 3.1 is presented suggests that there were no real trends between specific total lipid production rates and the tested parameters. Yet, the title of section 3.2 states that product-to-product yield followed the same trends than specific total lipid production rates, meaning there were no trend either? This is really unclear.

Thank you. We referred here to the finding that especially the incubation time has a significant influence on the presented data, but no clear trend can be observed for the other tested parameters.

Lines 269-271: Instead of writing below 60 and almost 120 µmol C-mol-1, indicate actual values and errors.

Thank you. We added the mean values and the error.

Lines 292-293: 39 and 49 % are averaged proportions of tetraethers. Indicate the deviations across samples.

Thank you for that comment. We added the error for these values.

Line 297: Indicate the actual values for better comparison/understanding.

Thank you. Again, we think that mean values and errors would not make sense in that context. Here, we just refer to the visual impression that the tetraether bars in the V/V=const graph tend to be larger on average than those of the n/n=const graph.

Line 314: Where can these data be found? Do GTGT0a specific production rates correlate with any of the tested parameters?

Thank you. These data can be found in Figure 7. No, we don’t see a clear trend for the GTGT-0a specific production rates. As the amount of GTGT-0a is very low, we don’t want to overinterpret these data. We changed the figure captions accordingly.

Lines 334-340: It appears the degree of methylation is calculated here as the ratios GMGT-0b/GMGT-0a or GMGT-0c/GMGT-0a while, following the ring index (averaged number of cycles per molecule) calculations described by Schouten and colleagues, it would be more accurate to calculate this as follows: (1 x GMGT-0b + 2 x GMGT-0c)/(GMGT-0a + GMGT-0b + GMGT-0c). Again, where can the values be found so that comparison could be made more easily. A plot of methylation degrees across conditions could also be interesting to visualize.

Thank you. We modified the calculations according to your suggestion. Furthermore, we visualized the results in a new figure and added the observations to the text.

Lines 358-359: This seems true only in very specific settings, i.e., small volumes where it is not even true for all the experiments in small volumes (see for instance 1.1 bar, 102.65 and 82.82 h are equivalent). The same applies for lines 360-361: there are some examples of n/n constant cultures where higher pressures yielded higher ODend and vice versa for V/V constant.

Thank you. As stated in the specific line, this could be only observed in n/n=const. Experiments at small volumes and 1.1 bar in that extend. What can be seen is that at large volumes this trend might go into the other direction (as can be seen for the 82.82 h pressure series). We adapted the text regarding this observation: “However, this trend for n/n = const could get reversed when dealing with larger volumes, as the 82.82 h pressure series at large volumes might imply.”

Lines 383-385: What is this assumption based on? It is not that clear to me, especially considering the rather large error bars already present here with strictly controlled conditions.

Thank you. In this study many different closed batch set-ups were used (n = 4 and N = 2), where gas was used as substrate. Such an ambitious experimental set-up is unprecedented and reveals that many experimental conditions in scientific studies might overlook that under only marginally varying growth conditions the results in repeated (N =2) experiments might obtain a different outcome. Although we did everything to repeat our experiments perfectly. Therefore, as one of the clear outcomes of our study, we suggest optimization and improvement avenues to perform studies in a repeatable and reproducible way regarding lipid production analysis. We wanted to point towards this fact.

Lines 385-386: Again, I do not understand why. Cultures at V/V constant often showed higher lipid production rates and product-to-product yields than cultures at n/n, and are thus, in my opinion, more promising in the prospect of upscaling cultures for lipid production.

Thank you. Yes, we agree that the production rate tends to be slightly higher at V/V=const, but at n/n=const the diversity is higher which makes it more interesting in our opinion.

Line 394: How does this compare to previous studies? Could this be even more increased, for instance by shaking the cultures?

Thank you. Yes, this would be definitely increased by shaking the cultures as the mixing rate would be significantly higher. That’s the reason why the cultures in our study where incubated in a water bath under continuous shaking. Based on our knowledge other studies with such an experimental set-up were not performed before.

Lines 434-438: Please reference this (for instance, Salvador-Castell et al. 2019, 10.3390/ijms20184434). This is also the first mention to squalene in the manuscript and I do not understand why it comes this late if it can be such a valuable compound. Does M. marburgensis produce squalene? Again, working with CL here probably degraded squalene and other such hydrocarbon derivatives preventing from understanding how the tested parameters control its production in M. marburgensis.

Thank you. We agree. This was clearly too far-fetched. We removed this paragraph from the manuscript.